# Global phenotypic profiling identifies a conserved actinobacterial cofactor for a bifunctional PBP-type cell wall synthase

Joel W Sher[1], Hoong Chuin Lim[1]*, Thomas G Bernhardt[1,2]*

[1]Department of Microbiology, Harvard Medical School, Boston, United States;
[2]Howard Hughes Medical Institute, Chevy Chase, United States

**Abstract** Members of the *Corynebacterineae* suborder of Actinobacteria have a unique cell surface architecture and, unlike most well-studied bacteria, grow by tip-extension. To investigate the distinct morphogenic mechanisms shared by these organisms, we performed a genome-wide phenotypic profiling analysis using *Corynebacterium glutamicum* as a model. A high-density transposon mutagenized library was challenged with a panel of antibiotics and other stresses. The fitness of mutants in each gene under each condition was then assessed by transposon-sequencing. Clustering of the resulting phenotypic fingerprints revealed a role for several genes of previously unknown function in surface biogenesis. Further analysis identified CofA (Cgp_0016) as an interaction partner of the peptidoglycan synthase PBP1a that promotes its stable accumulation at sites of polar growth. The related *Mycobacterium tuberculosis* proteins were also found to interact, highlighting the utility of our dataset for uncovering conserved principles of morphogenesis for this clinically relevant bacterial suborder.

*For correspondence:
hoongchuin.lim@gmail.com (HCL);
thomas_bernhardt@hms.harvard.
edu (TGB)

Competing interests: The authors declare that no competing interests exist.

## Introduction

Several medically important pathogens belong to the Corynebacterineae suborder of Actinobacteria, including *Corynebacterium diphtheriae* and *Mycobacterium tuberculosis* (*Mtb*). Like most bacteria, these organisms surround their cytoplasmic membrane with an essential cell wall matrix made of the heteropolymer peptidoglycan (PG). However, they uniquely modify the PG layer with an additional polysaccharide made of arabinan and galactan chains (*Kieser and Rubin, 2014*; *Alderwick et al., 2015*; *Daffé and Marrakchi, 2019*). This arabinogalactan (AG) layer is further modified by glycolipids called mycolic acids, forming a second membrane that is thought to function analogously to the outer membrane of Gram-negative bacteria (*Kieser and Rubin, 2014*; *Alderwick et al., 2015*; *Daffé and Marrakchi, 2019*). The overall envelope architecture of these bacteria is referred to as the mycolata cell envelope, and compounds that target its biogenesis are important components of current drug cocktails used to treat *Mtb* infections (*Alderwick et al., 2015*). Therefore, enhancing our understanding of the assembly mechanisms that construct the mycolata envelope has practical implications for anti-mycobacterial therapeutic discovery in addition to addressing a fundamental problem in microbiology.

Like all other Actinobacteria analyzed thus far, members of the Corynebacterineae grow by inserting new envelope material at their cell poles (*Flärdh, 2003*; *Daniel and Errington, 2003*). The mechanisms that govern tip growth in these organisms are ill-defined, but the DivIVA (Wag31) protein has long been known to play a key role in the process (*Flärdh, 2003*; *Letek et al., 2008*; *Nguyen et al., 2007*). This protein is thought to assemble into a cytoskeletal-like matrix lining the inner face of the cytoplasmic membrane at the cell poles (*Edwards and Errington, 1997*; *Ramamurthi and Losick, 2009*; *Lenarcic et al., 2009*; *Oliva et al., 2010*). Similar to FtsZ polymers that underly the cytokinetic ring, these DivIVA assemblies are believed to function by promoting the recruitment of cell envelope

synthases to the pole where they can promote surface elongation (*Kang et al., 2008*; *Melzer et al., 2018*). Indeed, both known classes of PG synthases have been found to localize to growing poles in several organisms (*Valbuena et al., 2007*; *Sieger et al., 2013*; *Sieger and Bramkamp, 2014*; *Hett et al., 2010*; *Kieser et al., 2015a*). These synthases include the bifunctional class A penicillin-binding proteins (aPBPs) (*Sauvage et al., 2008*) and the relatively recently characterized synthases composed of complexes formed between SEDS proteins and their class B PBP (bPBP) partners (*Meeske et al., 2016*; *Rohs et al., 2018*; *Taguchi et al., 2019*). Beyond a presumed DivIVA-requirement, it remains unclear how these PG synthases are recruited to the poles or how their activities are controlled and balanced with synthases involved in constructing the other envelope layers. Factors that mediate these important activities are likely to be encoded by genes of currently unknown function that are conserved among the Corynebacterineae.

Phenotypic profiling has proven to be a useful strategy to identify phenotypes for genes of unknown function to help uncover their biological activity. The method originally took advantage of the ordered knockout collections of yeast and *Escherichia coli* (*Nichols et al., 2011*; *Hillenmeyer et al., 2008*). Profiles were generated by replica-plating the libraries on agar containing different drugs or other stresses and the fitness of each mutant under each condition was assessed based on measurements of colony size. Similar approaches utilizing transposon-sequencing have recently been employed to generate profiles for several bacterial species (*Wetmore et al., 2015*; *Price et al., 2018*), but an extensive analysis has not yet been carried out in the Corynebacterineae. Therefore, to better understand cell envelope assembly and polar growth in these organisms, we performed a global phenotypic profiling analysis of the model bacterium *Corynebacterium glutamicum* (*Cglu*). For the analysis, we used our recently generated high-density transposon mutant library of *Cglu* (*Lim et al., 2019*) and challenged it with a panel of antibiotics and other stresses. The fitness of mutants in each gene under each condition was then assessed by comparing the change in the proportion of mapped transposon insertions within a gene following ten generations of growth under a given condition. Clustering of the resulting phenotypic fingerprints for each gene revealed a role in surface biogenesis for several genes of previously unknown function. Further analysis of one such gene identified CofA (Cgp_0016) as a specific interaction partner of an aPBP-type PG synthase called PBP1a. CofA was shown localize to the cell pole and to be required for PBP1a to stably accumulate at these sites. Furthermore, we found that cognate CofA-PBP proteins from *Mycobacterium tuberculosis* and a pathogenic corynebacterium also participate in specific interactions. Thus, our overall results identify a conserved new component of the polar growth machinery within the Corynebacterineae and highlight the utility of our phenotypic profiling dataset for uncovering common principles of morphogenesis for this important group of bacteria.

## Results

### Phenotypic profiling of a high-density *Cglu* transposon library

To generate phenotypic profiles, a *Cglu* transposon library of approximately 200,000 unique insertions was grown for eleven generations in the presence of drug or under a stress condition. For drug treatments, initial trials revealed that the best transposon-sequencing results were obtained when a drug concentration yielding a mild but observable decrease in growth rate was used. Typically these concentrations were 1/4 to 1/2 the measured minimal inhibitory concentration (MIC) for each drug (*Figure 1—figure supplement 1A* and *Supplementary file 1*). In several cases, two different treatment concentrations of a particular drug were used in the analysis. All total, 40 different growth conditions were analyzed with the collection of drugs used spanning all major targets, including PG and AG biogenesis, DNA replication, transcription, and translation (*Figure 1A* and *Supplementary file 1*).

Following growth in each condition, genomic DNA was isolated from the cultures and transposon insertion profiles were analyzed by sequencing. Based on the results, a fitness score for mutants in each gene, referred for simplicity as gene fitness, was calculated by comparing the proportion of transposon reads mapped in the gene following one generation of growth in the absence of treatment relative to those mapped after eleven generations of growth in the treatment condition (see Materials and methods). Scores below 1.0 indicate reduced fitness relative to the population, whereas scores greater than 1.0 indicate greater fitness. Replicates of untreated cultures resulted in

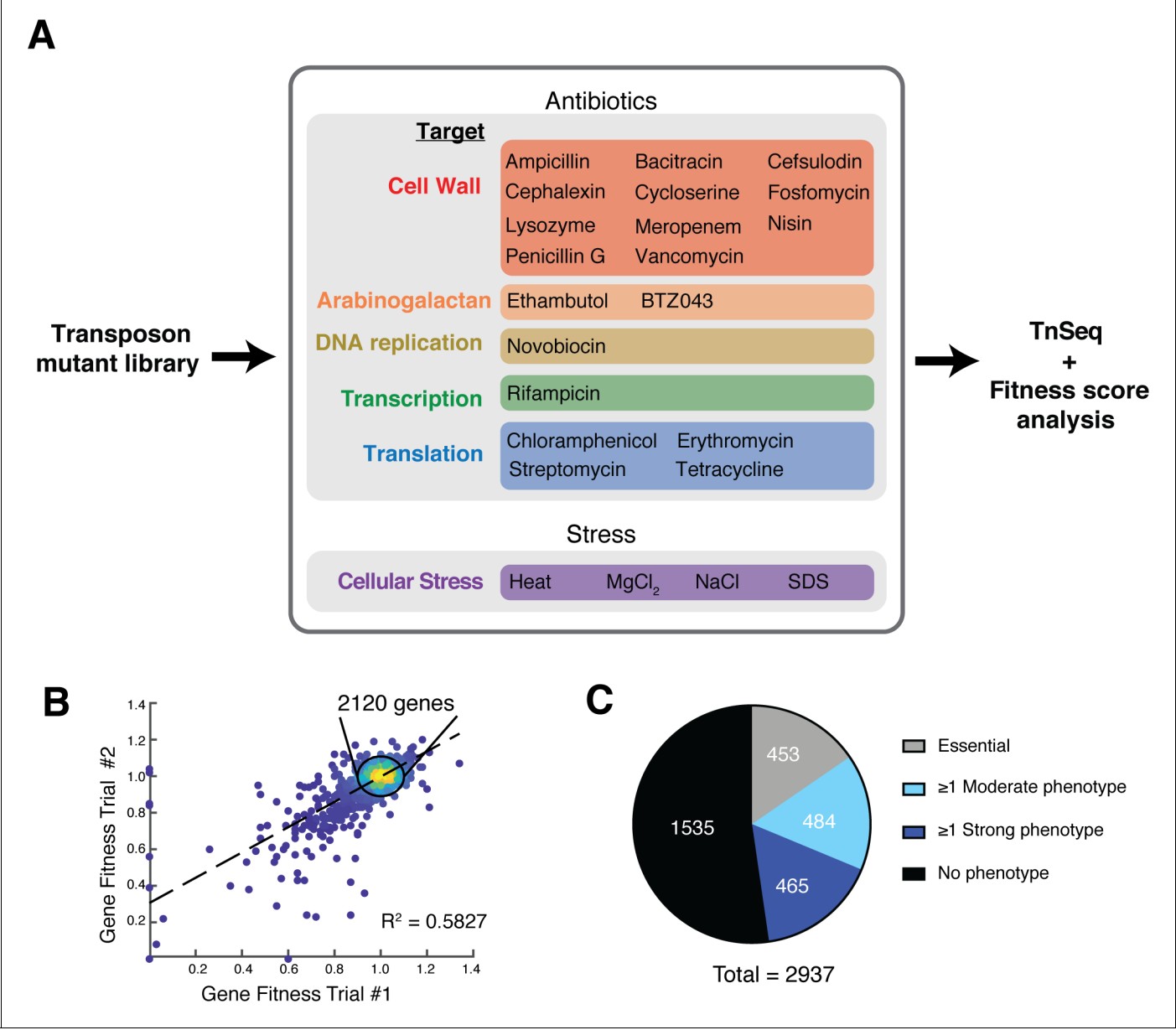

**Figure 1.** Phenotypic profiling of a *Corynebacterium glutamicum* transposon mutant library. (A) Overview of the phenotypic profiling procedure. A transposon mutagenized library of *Cglu* MB001 was exposed to sub-MIC concentrations of the indicated antibiotics or to the listed stress condition for 11 generations prior to transposon sequencing analysis and the calculation of fitness scores for mutants in each gene under each condition. Several of the antibiotics were tested at two different concentrations such that a total of 40 different growth conditions were surveyed (see *Supplementary file 1*). (B) Scatterplot highlighting the reproducibility of the analysis for duplicate samples grown in the absence of drug. The calculated fitness scores for each gene in the two replicates are plotted. Scores were calculated by comparing the proportion of total transposon reads for each gene in the untreated samples grown for 11 generations relative to the reads mapped for the input library. (C) Pie chart summarizing results from the profiling analysis. Depicted are essential genes (453, gray), genes that displayed a strong phenotype (fitness value below 0.75 or above 1.25) in at least one condition (465, dark blue), genes that displayed a moderate phenotype (fitness value below 0.9 or above 1.1) in at least one condition (484, light blue) and genes that did not show a phenotype in any condition tested (1535, black).

The online version of this article includes the following figure supplement(s) for figure 1:

**Figure supplement 1.** Example growth curves for *Cglu* treated with selected drugs.

highly correlated fitness scores for each gene, with most genes yielding a fitness score near 1.0 as expected for such a comparison (*Figure 1B*). Overall, approximately 40% of all non-essential genes were found to have a moderate phenotype in at least one condition tested, indicated by a fitness score below 0.9 or above 1.1 (*Figure 1C*), with the vast majority of phenotypes observed being fitness defects.

Hierarchical clustering of the phenotypic profiles generated from the analysis was used to identify sets of genes with similar fingerprints that may function together in the same biological pathway (*Figure 2* and *Figure 2—source data 1*). As an indication that the clustering was accurately identifying factors with similar functions, genes encoding components of several characterized protein complexes were found to have highly correlated phenotypic signatures. For example, genes encoding the two cytochrome d oxidase subunits (*cydA* and *cydB*) and cytochrome transporter (*cydC* and *cydD*) cluster tightly together, primarily due to their hypersensitivity to the benzothiazinone BTZ043 (*Figure 2*). This finding is consistent with published literature demonstrating synergistic effects between electron transport chain inhibitors and benzothiazinones against *Mtb* (*Lechartier and Cole, 2015*). Additionally, genes encoding RipC, FtsE, and FtsX proteins that together form a complex required for proper cell wall remodeling at the division site (*Lim et al., 2019*; *Tsuge et al., 2008*; *Maeda et al., 2016*) were also found to have correlated profiles (*Figure 2*).

Another notable feature of the profiling analysis is that it successfully differentiated the biological function of genes from the same genetic locus. For example, genes in the putative *cgp_3163–3168* operon clustered into two distinct groups. Mutants in four of these genes (*cgp_3163, cgp_3165, cgp_3166,* and *cgp_3168*) have been associated with defects in trehalose mycolate transport across the cytoplasmic membrane (*Yamaryo-Botte et al., 2015*). Accordingly, they were all found to cluster

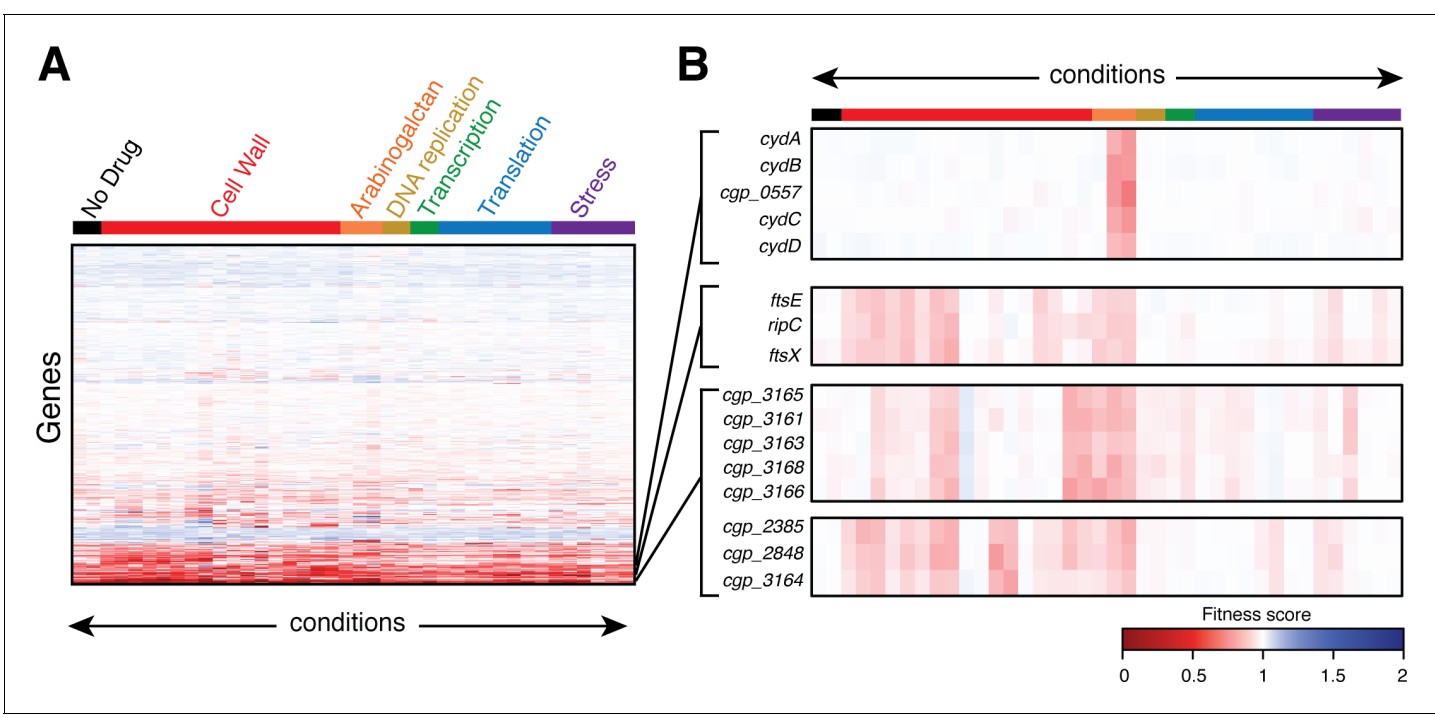

**Figure 2.** Clustering analysis of phenotypic profiles identifies genes with related functions. (**A**) Heatmap showing clustered phenotypic profiles for all non-essential genes in *Cglu*. The test conditions are oriented along the abscissa and are ordered by the stress or the physiological process affected by antibiotic treatment. The 2488 non-essential genes (excluding tRNAs, rRNAs, and transposons) are clustered along the ordinate with neighboring genes sharing similar fitness profiles across all conditions. The intensity of the red color indicates the magnitude of the fitness defect (dark red has a fitness value close to 0), white indicates a fitness value of 1, and blue color indicates a fitness advantages in a given condition. The full dataset is available in *Figure 2—source data 1*. (**B**) Expanded view of the profiles for select genes. See text for details.

The online version of this article includes the following source data and figure supplement(s) for figure 2:

**Source data 1.** Phenotypic profiling results.
**Figure supplement 1.** A genetic locus with a potential role in undecaprenol biogenesis or utilization.

together in the profiling analysis (*Figure 2*). Gene *cgp_3164* from this locus, on the other hand, was previously implicated in a different function. Mutants in this gene along with the unlinked *mptA* (*cgp_2385*) gene were found to have a defect in the elongation of the mannan backbone of membrane glycolipids (*Cashmore et al., 2017*). Consistent with this finding, *cgp_3164* clusters with *mptA* (*cgp_2385*) in our analysis and has a phenotypic profile distinct from its neighboring genes in the *cgp_3163–3168* locus (*Figure 2*). Therefore, the profiling and clustering results are not simply identifying functional groupings based on operonic organization.

In addition to properly correlating the function of factors known to work together within the same complex or pathway, the profiling data also identified possible roles in cell surface biogenesis for genes of previously unknown function. For example, the putative *cgp_3012–3020* operon stood out due to the specific hypersensitivity to bacitracin and vancomycin displayed by mutants in all of its genes except for *cgp_3015* (*Figure 2—figure supplement 1A–B*). This hypersensitivity was confirmed by deletion of the entire locus or individual genes (*cgp_3018* or *cgp_3019*) (*Figure 2—figure supplement 1C*). Also consistent with the phenotypic screen, these strains have a slight growth advantage on meropenem, but are not altered in their susceptibility to other drugs like ampicillin (*Figure 2—figure supplement 1B–C*). Vancomycin acts by directly binding the lipid II precursor for PG synthesis, which consists of the lipid carrier undecaprenol pyrophosphate (Und-PP) linked with the dissaccharide pentapeptide monomeric unit of PG (*Reynolds, 1989*; *Schneider and Sahl, 2010*). Bacitracin similarly targets the lipid stage of PG biogenesis by blocking the dephosphorylation of Und-PP products of PG glycan polymerases such that Und-P is not regenerated for use in the synthesis of lipid II, causing PG synthesis to be inhibited (*Schneider and Sahl, 2010*). Thus, the specific hypersensitivity of mutants in the *cgp_3012–3020* locus to drugs that interfere with lipid II biogenesis suggests that this large uncharacterized cluster of genes may be involved in undecaprenol synthesis or utilization to facilitate proper cell surface assembly in *Cglu*. Based on the overall accuracy of the functional connections made so far using the phenotypic profiling dataset, we anticipate that it will provide a useful resource for the discovery and characterization of new factors involved in cell surface biogenesis and other important biological processes within the Corynebacterineae.

## The profiles of *cgp_0016* and *ponA (cpg_0336)* are highly correlated

We continued mining the profiling data to identify genes of unknown function that had profiles that were highly correlated with those of factors known to play key roles in cell wall synthesis. This analysis peaked our interest in the gene of unknown function *cgp_0016,* which based on results presented below will henceforth be referred to as *cofA* (co-factor of PBP1a) (*Figure 3A*). The *cofA* gene encodes a small membrane protein of 114 amino acids with two predicted transmembrane domains and an N-in/C-in topology (*Figure 3B*). Its phenotypic profile clustered tightly with the *ponA* gene (*Figure 3A*) (correlation coefficient 0.91) encoding PBP1a, an aPBP-type synthase that helps build the PG layer. Like other aPBPs, PBP1a is a bitopic membrane protein with a small N-terminal cytoplasmic domain, a single transmembrane helix, and a large extracellular region (*Figure 3B*). The extracellular portion of the synthase contains both a glycosyltransferase (GTase, polymerase) domain that polymerizes lipid II into PG glycans and a transpeptidase (TPase) domain that forms the crosslinks between peptides of adjacent PG glycans in the wall matrix (*Figure 3B*). *Cglu* encodes a second aPBP called PBP1b, but its corresponding gene *ponB* (*cgp_3313*) had a phenotypic profile that was distinct from *cofA* and *ponA* (*Figure 3A*).

The high correlation of the *ponA* and *cofA* profiles suggested that CofA might be a cofactor specifically required for the function of PBP1a. Cofactors of aPBPs have been described in the Proteobacteria and Firmicutes (*Typas et al., 2010*; *Paradis-Bleau et al., 2010*; *Greene et al., 2018*; *Fenton et al., 2018*). However, no such cofactors have been identified as important for PG synthesis in the Corynebacterineae or other Actinobacteria. Given the critical roles played by these cofactors in the activation and/or control of PG synthesis by the aPBPs in other organisms, we decided to further investigate the connection between CofA and PBP1a in *Cglu*.

To validate the profiling results, deletions of *ponA* and *cofA* were constructed and the sensitivity of the resulting mutants to antibiotics was assessed. As expected, Δ*ponA* cells displayed hypersensitivity to ampicillin and meropenam relative to wild-type cells (*Figure 3C*). Cells deleted for *cofA* showed a similar sensitivity to these drugs, but in both cases the sensitivity defect was less severe than that of Δ*ponA* cells (*Figure 3C*). Double Δ*ponA* Δ*cofA* mutants displayed a drug sensitivity phenotype identical to the single Δ*ponA* mutant (*Figure 3C*). Thus, the mutant phenotypes of cells

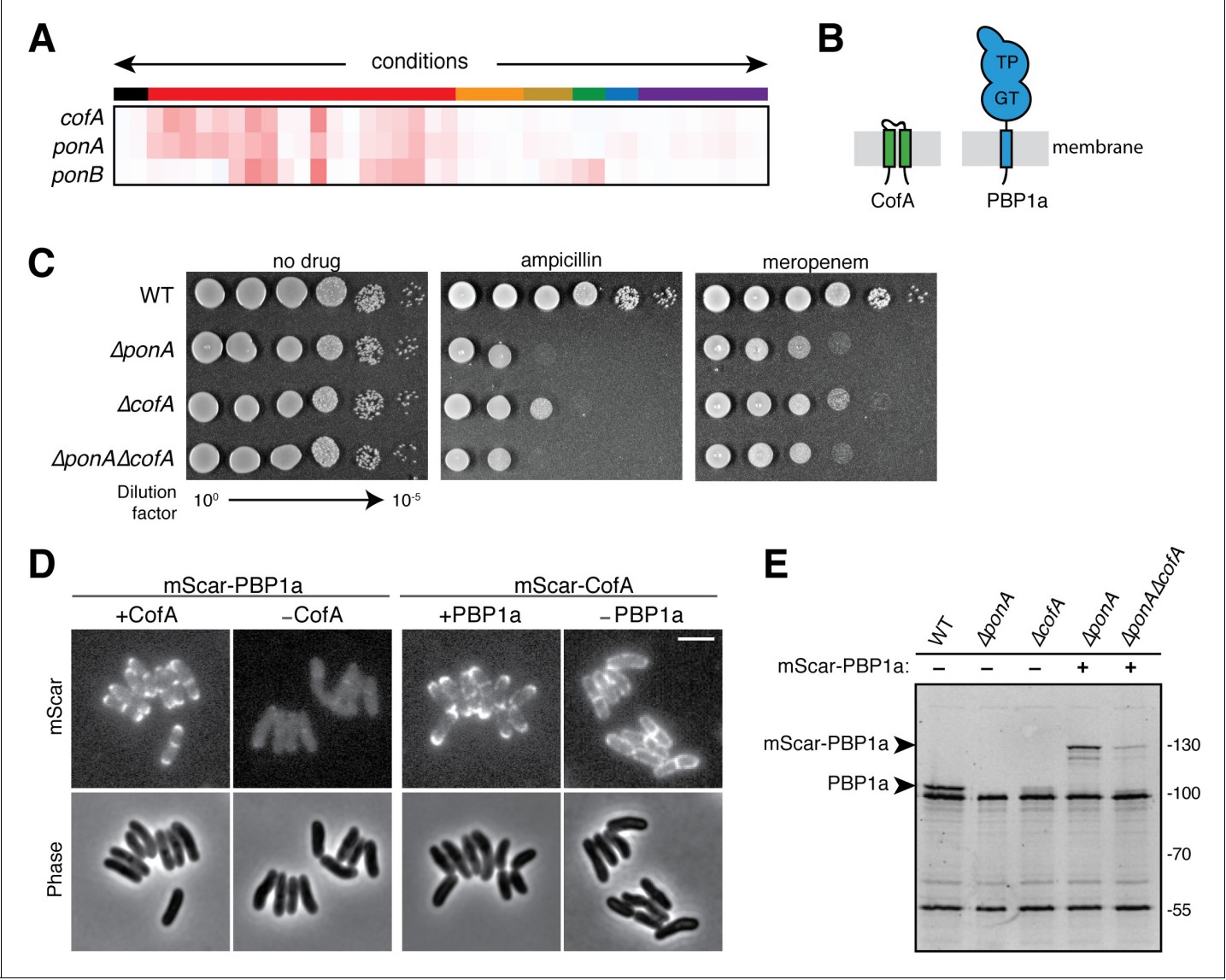

**Figure 3.** CofA is required for PBP1a accumulation. (**A**) Phenotypic profiles of *cofA* (*cgp_0016*), *ponA* (*cgp_0336*, encoding PBP1a), and *ponB* (*cgp_3313*, encoding PBP1b) displayed as in *Figure 2*. Note that *cofA* and *ponA* clustered tightly together in the analysis due to their similar profiles. Neither gene clustered near *ponB*, which is shown for reference. (**B**) Schematic showing the predicted membrane topology of CofA and PBP1a. (**C**) Cultures of wild-type *Cglu* and the indicated deletion mutants were grown, serially diluted, and plated as in *Figure 2—figure supplement 1*. The concentration of drugs used was 0.2 µg/mL ampicillin or 0.04 µg/mL meropenem as indicated. (**D**) Shown are mScarlet fluorescence (upper) and phase contrast (lower) micrographs of cells expressing the indicated fusion protein. Fusions were constitutively expressed from a construct integrated at the *attB1* site. Translation of the fusions was controlled by the theophylline (*riboE1*) riboswitch and was induced with 0.3 mM theophylline in each case. The fusions were produced in strains deleted for the corresponding native untagged protein. Cells from an overnight culture were diluted 1:1000 in BHI supplemented with 0.3 mM theophylline and then imaged on CGX2 agarose pads after growth for 5.5 hr at 30°C. The brightness for the two mScar-PBP1a micrographs is normalized to allow for direct comparison. Bar equals 3 µm. (**E**) Bocillin labeling of PBPs in wild-type and mutant strains. Overnight cultures of the indicated strains were diluted 1:200 in BHI and grown until they reached an $OD_{600}$ = 0.3. Cells were then treated with 10 µg/mL Bocillin-FL, and membrane fractions were isolated. Proteins (5 µg total) were then separated on a 10% SDS-PAGE gel and labeled bands were visualized using a Typhoon florescence scanner. Production of the mScar-PBP1a fusions was induced with 0.3 mM theophylline as for the microscopy analysis in panel D. Fluorescent band intensities for labeled PBP1a or mScar-PBP1a were quantified and normalized to the PBP2a band signal running just above 55 kDa. The PBP1a or mScar-PBP1a signal decreased by a factor of 5 in Δ*cofA* cells relative to the corresponding CofA⁺ strain.

The online version of this article includes the following figure supplement(s) for figure 3:

**Figure supplement 1.** Functionality of mScar fusion proteins and localization of catalytically defective PBP1a variants.

inactivated for PBP1a and CofA are indeed similar. However, the reduced severity of the phenotypes for CofA inactivation relative to PBP1a inactivation suggests that PBP1a retains some residual function in the absence of CofA. This phenotypic disparity is consistent with a model in which CofA promotes normal PBP1a function but is not absolutely required for its activity.

## CofA is required for the stable accumulation of PBP1a

PBP1a has been shown to localize to the cell pole where it is likely participating in polar surface growth (*Valbuena et al., 2007*). We reasoned that one potential function of CofA might be the recruitment of PBP1a to the growing pole. To investigate this possibility, we constructed N-terminal fusions of the red-fluorescent protein mScarlet (mScar) to both PBP1a and CofA. Both fusions were functional as they complemented the ampicillin hypersensitivity phenotype of the corresponding deletion allele (*Figure 3—figure supplement 1A*). As expected based on prior results, mScar-PBP1a displayed a prominent polar localization pattern with additional signal at developing division septa (*Figure 3D* and *Figure 3—figure supplement 1D*). This localization pattern did not require PBP1a activity as mutant fusions inactivated for polymerase (GTase), crosslinking (TPase), or both activities retained their polar recruitment signal (*Figure 3—figure supplement 1B–C*). mScar-CofA showed a very similar localization pattern (*Figure 3D* and *Figure 3—figure supplement 1D*). Notably, the polar localization pattern of mScar-CofA was lost in Δ*ponA* cells (*Figure 3D* and *Figure 3—figure supplement 1D*). The fusion instead displayed a peripheral localization signal typical of a delocalized membrane protein. In stark contrast, when mScar-PBP1a localization was assessed in Δ*cofA* cells, the fluorescent signal observed was dramatically reduced relative to cells producing CofA (*Figure 3D* and *Figure 3—figure supplement 1D*). This result suggested that PBP1a may not stably accumulate in the absence of CofA. To investigate this possibility, we used the fluorescent penicillin derivative called Bocillin to monitor PBP1a levels in wild-type and Δ*cofA* cells.

Like all beta-lactams, Bocillin covalently modifies the TP active site of PBPs. It can therefore be used label cellular PBPs to track their abundance in membrane extracts following protein denaturation, and separation by SDS-PAGE. Labeling of wild-type cells yielded three bands of high-intensity (*Figure 3E*). The top band was identified as PBP1a due to its absence in the profile of labeled Δ*ponA* cells (*Figure 3E*). As expected from the fluorescent microscopy analysis, the intensity of the PBP1a band was dramatically reduced in samples from Δ*cofA* cells (*Figure 3E*). Similarly, when Δ*ponA* cells expressing the mScar-PBP1a fusion were labeled, the PBP1a band was shifted to a higher molecular weight corresponding to the fusion, and this band was significantly reduced in intensity when CofA was inactivated (*Figure 3E*). Given that the fusion protein was produced from a heterologous promoter, the effect of CofA is unlikely to be at the level of *ponA* transcription. Rather, we infer from the data that CofA is most likely required for the stable accumulation of PBP1a.

## CofA directly interacts with the transmembrane domain of PBP1a

The accumulation defect of PBP1a in cells lacking CofA suggested that the two proteins might interact and that this interaction is required to stabilize PBP1a. To assess their potential interaction, we used the recently developed POLAR (PopZ-Linked Apical Recruitment) two-hybrid assay in *E. coli* (*Lim and Bernhardt, 2019*). For this method, a bait protein is fused to GFP and a peptide that recruits the fusion to polarly-localized assemblies of the *Caulobacter cresentus* PopZ protein. The prey protein is expressed as an mScar fusion, and interaction with the bait is assessed based on whether or not the prey is recruited to the polar PopZ assembly. With this assay, CofA was found to interact with PBP1a but not a control transmembrane domain fusion or a fusion to PBP1b, the other aPBP encoded by *Cglu* (*Figure 4A*). Additionally, CofA was found to self interact (*Figure 4B*). Notably, we also observed that the transmembrane domain of PBP1a was necessary and sufficient for a positive CofA-PBP1a interaction in the POLAR assay (*Figure 4C*). We thus conclude that CofA interacts directly and specifically with PBP1a via its transmembrane domain.

## The CofA-PBP1a interaction is required for the stable accumulation of PBP1a at the pole

We next tested whether the transmembrane domain of PBP1a was responsible for its failure to accumulate in the absence of CofA. To do so, we replaced the native PBP1a transmembrane domain with the corresponding domain from *E. coli* PBP1a in the context of the mScar fusion. Bocillin

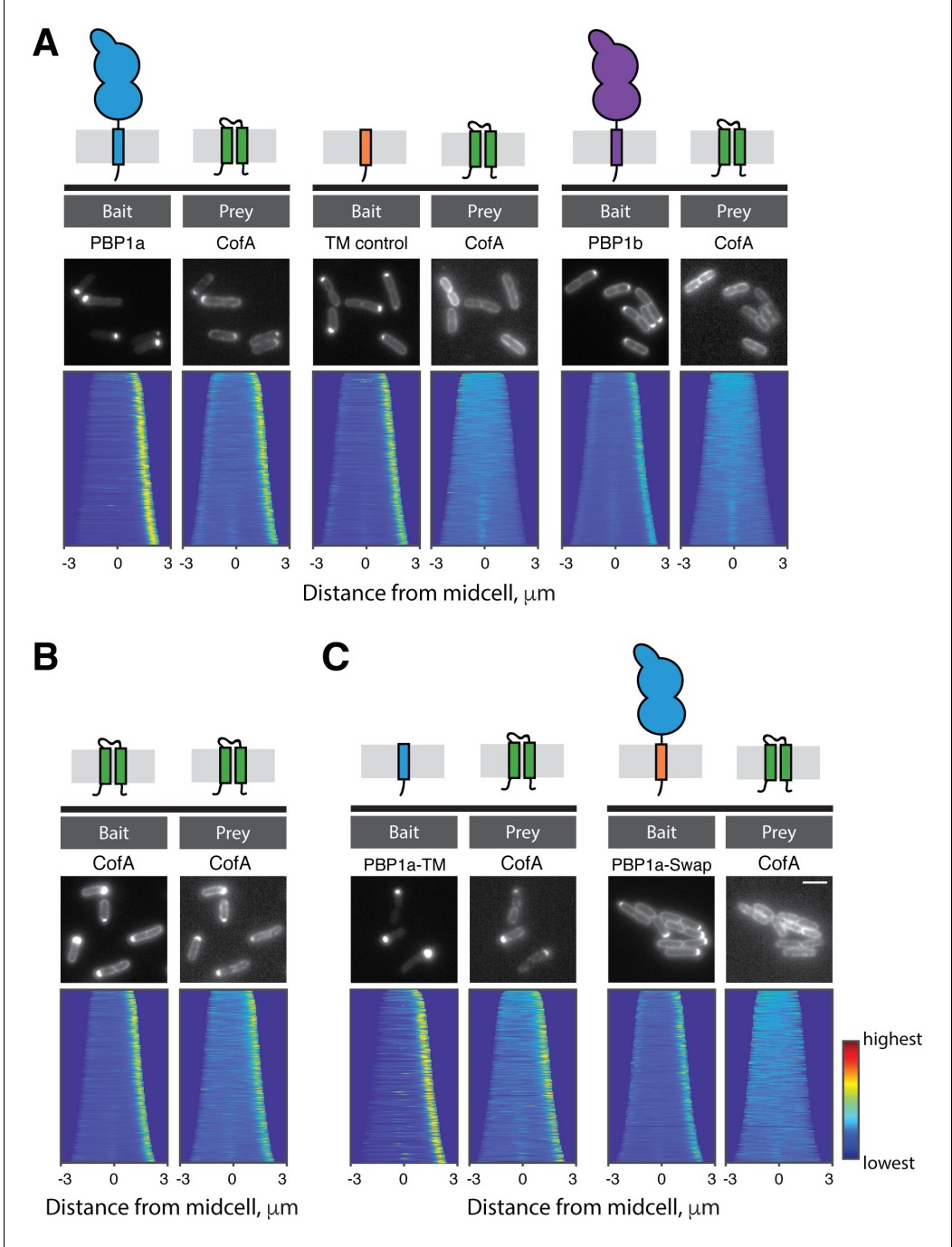

**Figure 4.** CofA specifically interacts with PBP1a. Shown are results from the POLAR two-hybrid assay with proteins expressed in *E. coli* cells. Bait proteins were fused with GFP and the H3H4 peptide to target them to polar assemblies of the PopZ protein. Prey proteins were expressed as mScar fusions. Schematics of the proteins or protein domains used as bait or prey are shown above the micrographs for reference. In each panel, representative fluorescence images of *E. coli* cells expressing the indicated bait and prey proteins are shown. Below these images are demographs that reflect protein localization throughout a population of cells. For the demographs the distribution of fluorescence across at least 250 cells was quantified. The resulting heatmaps of fluorescence intensity for each cell were then arranged according to cell length and stacked to generate the demograph. A custom-written MATLAB script was used to orient the cells such that the cell pole with the higher bait fluorescence was located on the right of the demograph (available at https://github.com/jsher-Bernhardtlab/cofA; *Sher, 2020*; copy archived at https://github.com/elifesciences-publications/cofA). (**A**) (left) GFP-PBP1a bait with mScar-CofA prey. (middle) Control transmembrane domain fused to GFP (GFP-TM) as bait with mScar-CofA prey. (right) GFP-PBP1b bait with mScar-CofA prey. (**B**) GFP-CofA bait with mScar-CofA prey. (**C**) (left) GFP fused to the tramsmembrane domain of PBP1a (GFP-PBP1a-TM) with mScar-CofA prey. (right) GFP-PBP1a with a heterologous transmembrane domain (GFP-PBP1a-Swap) with mScar-CofA prey.

labeling indicated that unlike wild-type mScar-PBP1a, this fusion protein (mScar-$^{Ec}$TM-PBP1a) accumulated in ΔcofA cells to levels approaching that of the wild-type mScar-PBP1a fusion in CofA$^+$ cells (*Figure 5A*). Although accumulation of PBP1a was restored by the transmembrane domain swap, the hybrid fusion was only partially functional in restoring normal levels of sensitivity to ampicillin to ΔponA *or* ΔponA ΔcofA cells (*Figure 5B*). Moreover, the hybrid fusion was not recruited to the cell

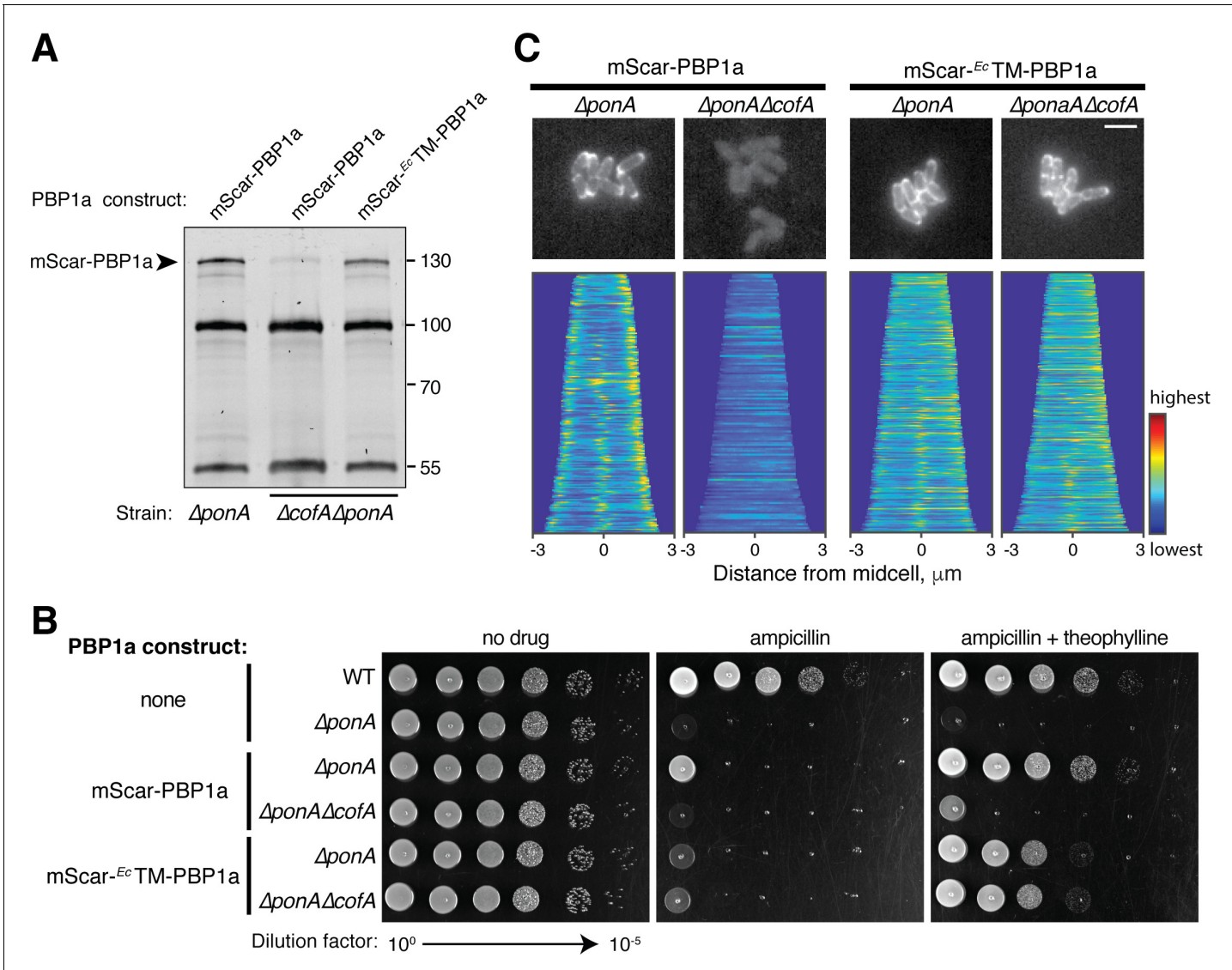

**Figure 5.** Interaction with CofA is required for polar localization of PBP1a. (A) Bocillin labeling of mScar-PBP1a in the indicated strains. Production of the mScar fusions was induced with 0.3 mM theophylline as in *Figure 3*. For this gel, 7.5 μg of total protein was loaded for each sample. mScar-$^{Ec}$TM-PBP1a refers to the PBP1a derivative in which the transmembrane domain of *E. coli* PBP1a was used to replace the corresponding domain of native *Cglu* PBP1a. Fluorescent band intensities for labeled mScar-PBP1a derivatives was performed as in *Figure 3E*. The mScar-PBP1a band decreased in intensity by a factor of 8 in ΔcofA cells relative to the corresponding CofA$^+$ strain, whereas the mScar-$^{Ec}$TM-PBP1a in ΔcofA cells was present at 74% of the mScar-PBP1a level in CofA$^+$ cells. (B) Cultures of the indicated strains encoding mScar-PBP1a, mScar-$^{Ec}$TM-PBP1a, or no fusion as indicated were grown and plated as in *Figure 2—figure supplement 1*. Plates contained 0.3 μg/mL ampicillin with or without 0.3 mM theophylline to induce the production of the PBP1a fusions as indicated. (C) (top) Shown are representative micrographs of ΔponA or ΔponA ΔcofA cells producing the indicated mScar-PBP1a or mScar-$^{Ec}$TM-PBP1a fusion, which was induced upon addition of 0.3 mM theophylline. Cells were imaged on CGX2 with supplements agarose pads. (bottom) Demographs showing fluorescence distribution of the corresponding mScar fusions throughout a population of cells. At least 150 cells were analyzed for each demograph. Bar equals 3 μm.

The online version of this article includes the following figure supplement(s) for figure 5:

**Figure supplement 1.** CofA does not interact with PBP1a with a heterologous transmembrane domain.

poles like wild-type mScar-PBP1a, but instead displayed a peripheral localization pattern consistent with a delocalized membrane protein (*Figure 5C*). Analysis by the POLAR assay also found that the $^{Ec}$TM-PBP1a hybrid failed to interact with CofA (*Figure 5—figure supplement 1*). The overall results with the hybrid PBP1a fusion are therefore consistent with a model in which the CofA-PBP1a interaction is needed for the stable accumulation of the complex at the cell pole.

## The CofA-PBP1a interaction is conserved among the Corynebacterineae

The CofA protein from *Cglu*, including its two transmembrane domains, consists almost entirely of the domain of unknown function designated DUF3566. Proteins with this domain are found throughout the Actinobacteria, and as in *Cglu*, the *cofA* sequence is commonly located in the same genetic context, directly downstream of DNA gyrase and upstream of tRNA gene(s) (*Figure 6A–B*). Some of the CofA paralogs have extended N-terminal cytoplasmic domains that may provide added functionality to the protein (*Figure 6B*). We tested the ability of CofA proteins from the opportunistic pathogen *Corynebacterium jeikium* and *Mtb* for their ability to interact with aPBPs using the POLAR assay. The *C. jeikium* CofA ($^{Cj}$CofA) was found to interact with the transmembrane domain of *C. jeikium* PBP1a ($^{Cj}$PBP1a-TM), but not the corresponding domain of its PBP1b ortholog ($^{Cj}$PBP1b-TM) (*Figure 6C*). Using the full-length CofA from *Mtb* ($^{Mtb}$CofA-FL) as the prey, a weak interaction signal was detected with the transmembrane domain of $^{Mtb}$PonA2, the paralog of *Cglu* PBP1a, but not the transmembrane domain of $^{Mtb}$PonA1, the paralog of *Cglu* PBP1b (*Figure 6—figure supplement 1*). For reasons that are currently unclear, the specific interaction signal between $^{Mtb}$CofA and the transmembrane domain of $^{Mtb}$PonA1 was greatly improved when the N-terminal extension of CofA was deleted ($^{Mtb}$CofA-ΔN) (*Figure 6D*). We thus conclude that the CofA-PBP1a interaction is conserved among the Corynebacterineae and most likely also throughout the Actinobacteria.

## Discussion

The mechanisms underlying envelope biogenesis, polar cell growth, and cell division remain poorly understood for the Corynebacterineae suborder and the Actinobacteria phylum in general. Genes of currently unknown function are likely to encode factors that play key roles in these major growth processes. To uncover the functions of these factors and gain new insights into the cell cycle events they participate in, growth phenotypes for mutants of their corresponding genes and novel genetic handles are needed. We therefore performed a global phenotypic profiling analysis of *Cglu*, a model member of the Corynebacterineae. As with similar analyses in other bacterial systems (*Nichols et al., 2011*), one of the major utilities of this genetic resource is the ability to identify groups of genes that are likely to participate in the same biological pathway or process. These groupings are especially useful when they connect genes of unknown function with genes that have well-established activities. Several examples were presented to demonstrate how the dataset reported here successfully identifies genes with similar function or that participate in the same protein complex, including genes associated with the transport of trehalose mycolate components of the mycolata outer membrane. Furthermore, the identification of CofA as a new component of the PG synthesis machinery showed how the dataset can be used for discovering new components of the polar growth machinery.

The phenotypic profiling data initially suggested that CofA might be a cofactor required for the function of the aPBP-type PG synthase PBP1a, and our subsequent analysis confirmed this hypothesis. Cofactors of aPBPs were originally discovered in the Gram-negative bacterium *Escherichia coli* (*Typas et al., 2010*; *Paradis-Bleau et al., 2010*). In this organism, each of the two major aPBPs, $^{Ec}$PBP1a and $^{Ec}$PBP1b, associates with a cognate outer membrane lipoprotein that is required for its in vivo function. Additionally, biochemical studies indicate that the $^{Ec}$LpoA cofactor primarily stimulates the crosslinking activity of its $^{Ec}$PBP1a partner, whereas the glycan polymerase activity of $^{Ec}$PBP1b is activated by its cofactor $^{Ec}$LpoB (*Typas et al., 2010*; *Paradis-Bleau et al., 2010*; *Egan et al., 2014*; *Lupoli et al., 2014*). Genetic studies in related Gram-negative bacteria indicate that their aPBPs also require similar outer membrane lipoproteins for activity (*Dörr et al., 2014*; *Yin et al., 2015*). The exception is *Pseudomonas aeruginosa* and its relatives where a LpoB ortholog is missing and $^{Pa}$PBP1b instead relies on the unrelated membrane lipoprotein LpoP for its function (*Greene et al., 2018*). Thus, aPBP cofactors are diverse and are thought to have evolved relatively recently (*Typas et al., 2010*; *Typas et al., 2012*). Accordingly, studies of the Gram-positive pathogen *Streptococcus pneumoniae*, which lacks an outer membrane, have identified distinct set of aPBP

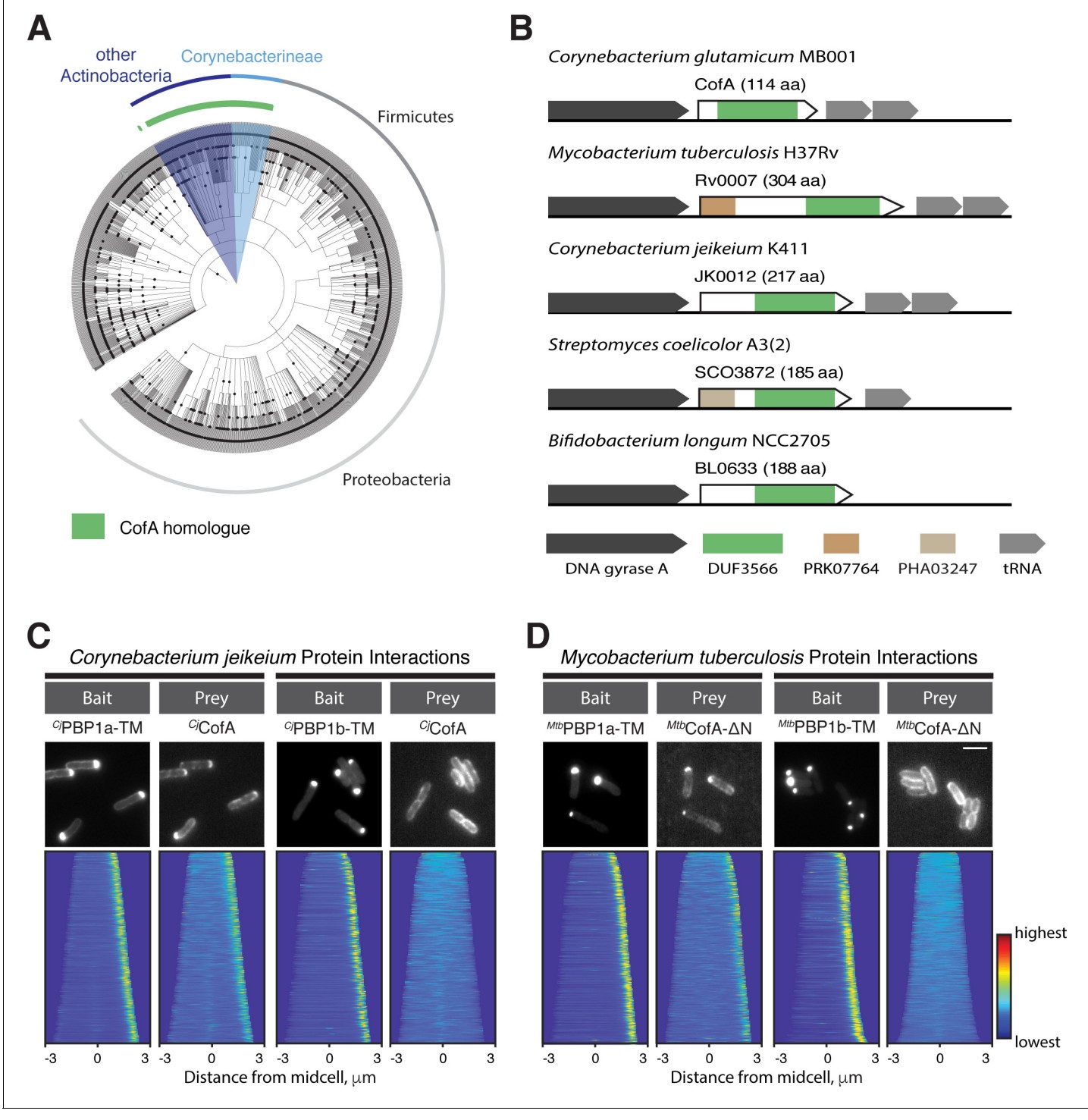

**Figure 6.** The CofA-PBP1a interaction is conserved. (**A**) Phylogenetic tree showing the distribution of CofA-like proteins containing the DUF3566 domain. (**B**) Schematics showing the genomic organization of loci encoding *cofA*-like genes in representative bacteria. The domain predictions are based on the NCBI conserved domain database. Not drawn to scale. (**C–D**) POLAR two-hybrid assay assessing the interaction of CofA paralogs from *Corynebacterium jeikeium K411* (**C**) or *Mycobacterium tuberculosis H37Rv* (**D**) with PBPs from these organisms. Results are displayed as in *Figure 4*. (**C**) (left) GFP fused to the transmembrane domain of *C. jeikeium* PBP1a (GFP-$^{Cj}$PBP1a-TM) with mScar-$^{Cj}$CofA prey. Reading frames used were JK1977 (residues 1–44) and JK0012, respectively. (right) GFP fused to the transmembrane domain of *C. jeikeium* PBP1b (GFP-$^{Cj}$PBP1b-TM) with mScar-$^{Cj}$CofA prey. Reading frames for PBP1b used was JK2069 (residue 185–235). Bar equals 3 μm. (**D**) POLAR two-hybrid assay results with *Mtb* proteins. (left) GFP fused to the transmembrane domain (residues 1–41) of *Mtb* PonA2 (GFP-$^{Mtb}$PonA2-TM) with mScar-$^{Mtb}$CofA-ΔN prey. Note PonA2 (Rv3682) is the *Mtb*

*Figure 6 continued on next page*

*Figure 6 continued*

ortholog of *Cglu* PBP1a. The CofA fusion used is deleted for the N-terminal extension (residues 1–190) found on the *Mtb* CofA sequence (Rv0007). (right) GFP fused to the transmembrane domain (residues 120–167) of *Mtb* PonA1 (GFP-*Mtb*PonA1-TM) with mScar-*Mtb*CofA-ΔN prey. Note PonA1 (Rv0050) is the *Mtb* ortholog of *Cglu* PBP1b. Bar equals 3 μm.

The online version of this article includes the following figure supplement(s) for figure 6:

**Figure supplement 1.** Interaction of full length *Mtb*CofA with PonA2.

partners that are thought to promote proper PG synthesis. The polytopic membrane protein CozE was identified as a factor that is essential for controlling the subcellular localization of *SP*PBP1a, whereas the bitopic membrane protein MacP was found to be required for *SP*PBP2a in a manner that requires phosphorylation by the Ser-Thr kinase StkP (*Fenton et al., 2018*; *Fenton et al., 2017*). The effect of these *S. pneumoniae* factors on the biochemical activities of their cognate aPBP remains to be determined.

To our knowledge, cofactors for aPBPs in the Actinobacteria have not been previously described. CofA has many of the expected hallmarks of such a cofactor. Its loss of function phenocopies the inactivation its cognate aPBP, and it interacts specifically with this synthase. What is unique about CofA compared to other aPBP cofactors characterized previously is that it is required for the stable accumulation of its partner enzyme. Because the accumulation defect of PBP1a in the absence of CofA is observed when the synthase gene is expressed from a heterologous promoter, it is unlikely to be due to problems with transcription of the *ponA* gene. Additionally, it is difficult to imagine how a small bitopic membrane protein would be required for the proper translation of the *ponA* reading frame. Instead, because components of membrane protein complexes are commonly found to depend on their partner proteins for stability (*Taura et al., 1993*), the most likely explanation for the CofA-dependence for PBP1a accumulation is that the synthase is subject to proteolytic degradation in the absence of its interaction partner. Whether CofA has additional roles in PBP1a function is not clear, but the observation that stabilized PBP1a variants that fail to interact with CofA also do not localize to the cell pole suggests a role for CofA in the recruitment of PBP1a to the polar growth zones. Further work will be required to determine how the CofA-PBP1a complex is localized to the poles, whether it includes additional proteins, and if CofA or other associated factors modulate the PG synthase activities of PBP1a. It also remains to be determined whether the other *Cglu* aPBP, PBP1b, also requires a cofactor for its function. No candidates were identified in the profiling analysis, but this absence of possible partners in the dataset may be because of redundancy in cofactor function or because the putative cofactor has additional activities beyond PBP1b activation that confound the phenotypic clustering.

Several lines of evidence suggest that the function of CofA as an aPBP cofactor is conserved among the Corynebacterineae and possibly among the Actinobacteria in general. Paralogs of CofA are found throughout the Actinobacteria with the corresponding gene located in a similar genetic context near the replication origin and downstream of a gene encoding a DNA gyrase. Furthermore, the POLAR two-hybrid analysis indicates that CofA proteins from both *Mtb* and *C. jeikium* also specifically interact with the PBP1a paralogs from their corresponding organism. Additional support for a conserved function for CofA is also found in high-throughput genetic datasets performed in *Mtb*. A Tn-seq analysis found that *Mtb*CofA (Rv0007) became less dispensable in cells inactivated for its non-cognate aPBP, PonA1 (*Kieser et al., 2015b*). Given the synthetic lethal relationship between PonA1 and PonA2 in *Mtb* (*Kieser et al., 2015b*), this result is consistent with CofA being required for PonA2 function. Similarly, a different Tn-Seq study found inactivation of *Mtb*CofA and PonA2 to both have suppressive genetic interactions with *rv1565c* (*DeJesus et al., 2017*), and a small scale phenotypic profiling screen in *Mtb* found that the phenotypes of *Mtb*CofA inactivation correlated with the inactivation of PonA2 (*Xu et al., 2017*). Thus, several lines of evidence indicate that *Mtb*CofA is likely to function as an aPBP cofactor in *Mtb*. Notably, *Mtb*CofA has a large N-terminal extension (190 a.a.) relative to the *Cglu* protein, and this domain has been found to be a target for phosphorylation (*Verma et al., 2017*). Additionally, removal of the N-terminal extension enhanced the interaction signal between *Mtb*CofA with the transmembrane domain of PonA2 observed in the POLAR assay. Therefore, PonA2 activity in *Mtb* might be controlled by phosphorylation of its cofactor just as PonA1 activity is thought to be modulated by direct phosphorylation (*Kieser et al., 2015a*).

In conclusion, we have identified a new cofactor required for the activity of an aPBP-type PG synthase that is conserved among the Actinobacteria. The localization of this factor along with its cognate aPBP at the cell pole suggests that these proteins form part of the cell elongation machinery in this important group of organisms. Further mining of the phenotypic profiling data provided in this study promises to reveal additional insights into the unique growth and division mechanism of these bacteria.

# Materials and methods

## Key resources table

| Reagent type (species) or resource | Designation | Source or reference | Identifiers | Additional information |
|---|---|---|---|---|
| Strain, strain background (*Escherichia coli*) | DH5α(λpir) | Gibco BRL | *F– hsdR17 deoR recA1 endA1 phoA supE44 thi-1 gyrA96 relA1 Δ(lacZYA-argF) U169 φ80dlac ZΔM15 ****add pir* | |
| Strain, strain background (*Escherichia coli*) | TB28 | (**Bernhardt and de Boer, 2003**) | MG1655 ΔlacIZYA::frt | |
| Strain, strain background (*Corynebacterium glutamicum*) | MB001 | (**Baumgart et al., 2013**) | ATCC 13032 ΔCGP1 (cg1507-cg1524) ΔCGP2 (cg1746-cg1752) ΔCGP3 (cg1890-cg2071) | |
| Strain, strain background (*Corynebacterium glutamicum*) | HL18 | This work | MB001 ΔponA | Bernhardt lab (MB001/pHCL86, see Materials and methods) |
| Strain, strain background (*Corynebacterium glutamicum*) | JS5 | This work | MB001 Δcgp_3012-cgp_3020 | Bernhardt lab (MB001/pJWS1, see Materials and methods) |
| Strain, strain background (*Corynebacterium glutamicum*) | JS6 | This work | MB001 Δcgp_3019 | Bernhardt lab (MB001/pJWS2, see Materials and methods) |
| Strain, strain background (*Corynebacterium glutamicum*) | JS7 | This work | MB001 Δcgp_3018 | Bernhardt lab (MB001/pJWS3, see Materials and methods) |
| Strain, strain background (*Corynebacterium glutamicum*) | JS8 | This work | MB001 Δcgp_0016 | Bernhardt lab (MB001/pJWS4, see Materials and methods) |
| Strain, strain background (*Corynebacterium glutamicum*) | JS10 | This work | MB001 Δcgp_0016 ΔponA | Bernhardt lab (JS8/pHCL86, see Materials and methods) |
| Recombinant DNA reagent | pJWS1 | This work | Kan^R, pCRD206 derivative containing an insert covering upstream and downstream of cgp_3012-cgp_3020. | Bernhardt lab (see **Supplementary file 2**) |
| Recombinant DNA reagent | pJWS2 | This work | Kan^R, pCRD206 derivative containing an insert covering upstream and downstream of cgp_3019. | Bernhardt lab (see **Supplementary file 2**) |

*Continued on next page*

*Continued*

| Reagent type (species) or resource | Designation | Source or reference | Identifiers | Additional information |
|---|---|---|---|---|
| Recombinant DNA reagent | pJWS3 | This work | Kan$^R$, pCRD206 derivative containing an insert covering upstream and downstream of *cgp_3018*. | Bernhardt lab (see *Supplementary file 2*) |
| Recombinant DNA reagent | pJWS4 | This work | Kan$^R$, pCRD206 derivative containing an insert covering upstream and downstream of *cgp_0016 (cofA)*. | Bernhardt lab (see *Supplementary file 2*) |
| Recombinant DNA reagent | pJWS18 | This work | Kan$^R$, pK-PIM derivative encoding P*sod-riboE1-mscar-cgp_0016 (cofA)* | Bernhardt lab (see *Supplementary file 2*) |
| Recombinant DNA reagent | pJWS19 | This work | Kan$^R$, pK-PIM derivative encoding P*sod-riboE1-mscar-cgp_0336 (ponA)* | Bernhardt lab (see *Supplementary file 2*) |
| Recombinant DNA reagent | pJWS97 | This work | Kan$^R$, pK-PIM derivative encoding P*sod-riboE1-mscar-*$^{C. glu}$*ponA* ($^{E. coli}$TM) | Bernhardt lab (see *Supplementary file 2*) |
| Recombinant DNA reagent | pJWS102 | This work | Kan$^R$, pK-PIM derivative encoding P*sod-riboE1-mscar-cgp_0336 (ponA)* GT- (E97A) | Bernhardt lab (see *Supplementary file 2*) |
| Recombinant DNA reagent | pJWS103 | This work | Kan$^R$, pK-PIM derivative encoding P*sod-riboE1-mscar-cgp_0336 (ponA)* TP- (S393A) | Bernhardt lab (see *Supplementary file 2*) |
| Recombinant DNA reagent | pJWS104 | This work | Kan$^R$, pK-PIM derivative encoding P*sod-riboE1-mscar-cgp_0336 (ponA)* GT- and TP- (E97A and S393A) | Bernhardt lab (see *Supplementary file 2*) |
| Recombinant DNA reagent | pHCL86 | (*Lim et al., 2019*) | Kan$^R$, pCRD206 derivative containing an insert covering upstream and downstream of *cgp_0336 (ponA)*. | |
| Recombinant DNA reagent | pHCL149 | (*Lim and Bernhardt, 2019*) | Cm$^R$, P$_{ara}$-*popZ-rbs-H3H4-msfGFPN-tm$_{ponB}$*. | |
| Recombinant DNA reagent | pHCL152 | (*Lim and Bernhardt, 2019*) | Tet$^R$, lacl-P$_{lac}$-*mscar*. | |
| Recombinant DNA reagent | pJWS29 | This work | Cm$^R$, *Para-popZ-rbs-H3H4-msfGFP-ponA* | Bernhardt lab (see *Supplementary file 2*) |
| Recombinant DNA reagent | pJWS41 | This work | Tet$^R$, lacl-P$_{lac}$-*mscar-cgp_0016* | Bernhardt lab (see *Supplementary file 2*) |
| Recombinant DNA reagent | pJWS70 | This work | Tet$^R$, lacl-P$_{lac}$-*mscar-jk0012* | Bernhardt lab (see *Supplementary file 2*) |
| Recombinant DNA reagent | pJWS73 | This work | Cm$^R$ P$_{ara}$-*popZ-rbs-H3H4-msfGFP-ponA$_{TM}$* | Bernhardt lab (see *Supplementary file 2*) |
| Recombinant DNA reagent | pJWS75 | This work | Cm$^R$, P$_{ara}$-*popZ-rbs-H3H4-msfGFP-ponA$_{Swap}$* | Bernhardt lab (see *Supplementary file 2*) |
| Recombinant DNA reagent | pJWS78 | This work | Cm$^R$, P$_{ara}$-*popZ-rbs-H3H4-jk1977$_{TM}$* | Bernhardt lab (see *Supplementary file 2*) |
| Recombinant DNA reagent | pJWS80 | This work | Cm$^R$, P$_{ara}$-*popZ-rbs-H3H4-msfGFP-ponB* | Bernhardt lab (see *Supplementary file 2*) |
| Recombinant DNA reagent | pJWS81 | This work | Cm$^R$, *Para-popZ-rbs-H3H4-ponA2$_{TM}$* | Bernhardt lab (see *Supplementary file 2*) |

*Continued on next page*

*Continued*

| Reagent type (species) or resource | Designation | Source or reference | Identifiers | Additional information |
|---|---|---|---|---|
| Recombinant DNA reagent | pJWS83 | This work | $Tet^R$, lacI-$P_{lac}$-mscar-rv0007$_{FL}$ | Bernhardt lab (see *Supplementary file 2*) |
| Recombinant DNA reagent | pJWS88 | This work | $Cm^R$, $P_{ara}$-popZ-rbs-H3H4-ponA1$_{TM}$ | Bernhardt lab (see *Supplementary file 2*) |
| Recombinant DNA reagent | pJWS90 | This work | $Cm^R$, $P_{ara}$-popZ-rbs-H3H4-jk2069$_{TM}$ | Bernhardt lab (see *Supplementary file 2*) |
| Recombinant DNA reagent | pJWS114 | This work | $Tet^R$, lacI-$P_{lac}$-mscar-rv0007$_{\Delta N}$ | Bernhardt lab (see *Supplementary file 2*) |
| Recombinant DNA reagent | pJWS119 | This work | $Cm^R$, $P_{ara}$-popZ-rbs-H3H4-$^{C. glu}$ponA ($^{E. coli}$TM) | Bernhardt lab (see *Supplementary file 2*) |
| Chemical compound | Bocillin | ThermoFisher Scientific | BOCILLIN FL Penicillin, Sodium Salt | |

## Media, bacterial strains, and plasmids

All *Cglu* strains used in the reported experiments are derivatives of MB001 (*Baumgart et al., 2013*). Unless mentioned otherwise, strains were grown in Brain Heart Infusion (BHI) medium that was supplemented with Kanamycin; Kan 15 µg/mL, when necessary. CGXII media (*Keilhauer et al., 1993*) supplemented with 30 mg/L thiamine, 30 mg/L calcium pantothenate, 200 mg/L nicotinamide and 0.2% casamino acids was used where indicated in figure legends. All *E. coli* strains used in the reported POLAR two-hybrid experiments are derivatives of TB28 (*Bernhardt and de Boer, 2004*) and were grown in LB (1% tryptone, 0.5% yeast extract, 0.5% NaCl). Whenever necessary, antibiotics for *E. coli* cultures were used at 25 (kanamycin; Kan), 15 (chloramphenicol; Cm), or 5 (tetracycline; Tet) µg/mL. All strains and plasmids used are listed in key resources table. Details for plasmid constructions are provided in *Supplementary file 2*.

## Transposon sequencing

The *Cglu* transposon library consisting of approximately 200,000 unique Tn5 transposon insertions was described previously (*Lim et al., 2019*). An inoculum of five million cells (25X coverage of the library) was grown for eleven generations to a final $OD_{600}$ of 0.5 in the presence of drug and/or stress conditions at 30 degrees. The concentration of drug used was determined empirically to be one sufficient to provide selection while not forming a bottleneck. Typically, the concentration used caused a growth defect of roughly 8–50% relative to untreated cells over the ten generation growth period. Following growth, transposon sequencing libraries were prepared as previously described (*Lim et al., 2019*). Briefly, genomic DNA was extracted from strains using the Wizard Genomic DNA Purification Kit (Promega) and cleaned using Genomic DNA Clean and Concentrator (Zymo). Genomic DNA was fragmented using a Qsonica Q800RS Sonicator for 12 min (using a 15 s on and 15 s off pulse cycle) at 20% amplitude. Fragmented DNA was purified with 1.8 × volume Agentcourt AMPure XP beads (Beckman Coulter, Inc) and eluded into 30 µl water. Purified fragmented DNA was then treated with terminal deoxynucleotidyl transferase (TdT; Promega) in a 20 µl reaction with 1 µL 9.5 mM dCTP/0.5 mM ddCTP, 4 µl 5 × TdT reaction buffer and 0.5 µl rTdT at 37°C for 1 hr, then at 75°C for 20 min. TdT-treated DNA was purified with a Performa DTR Gel Filtration Cartridge (EdgeBio). Purified, TdT-treated DNA was used as a template in a PCR reaction to amplify the transposon junctions using the Easy-A Hi-Fi Cloning System (Agilent Technologies). The primers used were: PolyG-1st-1 5'-GTGACTGGAGTTCAGACGTGTGCTCTTCCGATC TGGGGGGGGGGGGGGGGGG-3' and Tn5-1st-1 5'-ACCTGCAGGCATGCAAGCTTCAGGG-3'. A second nested PCR was next performed to further amplify the transposon junctions and append the sequencing barcode. The primers used were generic NEBNext Multiplex Oligos for Illumina (NEB) and: Tn5-2nd-1 5' AATGATACGGCGACCACCGAGATCTACACTCTTTTCAGGGTTGAGATGTGTA TAAGAGA-3'. The final product was run on a 2% agarose gel, and fragments ranging from 200 to 500 bp were gel purified using QIAquick Gel Extraction Kit (Qiagen). Libraries were sequenced at the Tufts University Core Facility on a HiSeq 2500 (Illumina) on a 1 × 50 single end run.

## Transposon sequencing and data analysis

Reads were trimmed using trimmomatic (Bolger et al., 2014) to remove adapter sequences, and mapped to the MB001 genome using bowtie 1.0.0 (Langmead et al., 2009). Only unique insertions with at least two reads were included in the downstream analysis. For each gene, we then compared the relative proportion of total transposon reads before and after growth in each condition. This allowed us to calculate the gene fitness for each gene in each condition using the equation: Fitness = $\ln(N_{t2}*2^{10}/N_{t1})/\ln((1-N_{t2})*2^{10}/(1-N_{t1}))$ for all non-essential Cglu genes [(van Opijnen and Camilli, 2013) and custom R scripts]. In order to reduce noise, we averaged transposon sequencing results from three replicates of our initial transposon library (called 1 g_A, 1 g_B and 1 g_C). We then employed hierarchical clustering (using heatmap.2 function from gplots package in R) to group genes behaving similarly into the heatmap shown in Figure 2a.

## Strain construction

For gene deletion in Cglu, we used the pCRD206 temperature-sensitive plasmid (Okibe et al., 2011). Briefly, the deletion allele with regions corresponding to approximately 750 bp upstream and downstream of the desired deletion were inserted into pCRD206. The resulting plasmid was transformed into the appropriate recipient strain. Transformants were selected and propagated on BHI agar supplemented with 15 µg/mL Kan at 25°C. To select for clones, a few fresh colonies were purified on a fresh BHI Kan plate, which was incubated at 30°C for 36 hr. One or two of the resulting Kan[R] colonies were grown in BHI liquid medium at 30°C for several hours. The culture was then spread on a BHI agarose plate supplemented with 10% sucrose, which was then incubated at 30°C. The resulting colonies were replica patched on BHI and BHI Kan plates to identify Kan-sensitive colonies. The deletion allele was finally confirmed by colony PCR. Plasmid integration at the attB1 site was performed using derivatives of the pK-PIM vector (Oram et al., 2007). Plasmids containing the desired expression construct were introduced into the recipient by direct transformation.

## Preparation of electrocompetent *C. glutamicum*

A stationary phase Cglu culture (10 ml) was diluted into 1 L BHIS (BHI + 91 g/L sorbitol) that was supplemented with 25 g glycine, 0.4 g isoniazid and 0.1% Tween 80 and incubated with shaking at 18°C. The culture was chilled on ice for 1 hr when the $OD_{600}$ reached 0.5 (typically in 16–18 hr). Cells were then collected by centrifugation at 4000 x g for 20 min. The pellet was washed once with 500 ml chilled 10% glycerol and three additional times with 100 ml chilled 10% glycerol. Cell density was adjusted to an $OD_{600}$ of 20 before use for electroporation.

## Electroporation of DNA

Approximately 100 ng of DNA was mixed with 100 µl of electrocompetent cells in a 1 mm electroporation cuvette (Genesee Scientific). The cells were then electroporated at 1.7kV using a MicroPulser electroporator (Bio-Rad). The cells were recovered in 1 ml BHIS, and then immediately heat-shocked for 6 mins at 46 degrees. The cells were then grown at 30 degrees for 1 hr before plating.

## Bocillin labeling

Bocillin labeling of PBPs was performed as described previously (Cho et al., 2016) with the following modifications. After incubation with Bocillin washing the cell pellets 3X with PBS (137 mM NaCl, 2.7 mM KCl, 8 mM $Na_2HPO_4$, and 2 mM $KH_2PO_4$), the pellets were frozen at −80°C. Cell pellets were then thawed, resuspended in 1 mL of PBS and lysed using a FastPrep-24 (MPBio). Cells were lysed using matrix B in 2 mL tubes for 40 s at 6 m/s. Cells were centrifuged for 3 min at 7,000 rpm to remove the undisrupted cells and matrix and the supernatant was transferred to a fresh tube. Membranes were then pelleted by ultracentrifugation at 100,000 × g for 20 min at 4°C. The membrane pellets were then washed with 1X PBS and resuspended in 60 µL 1X PBS. Resuspended samples were mixed with 60 µL 2X Laemmli sample buffer (100 mM Tris-Cl (pH 6.8), 4% (w/v) SDS, 0.2% (w/v) bromophenol blue, 20% (v/v) glycerol, 5% (v/v) β-mercaptoethanol) and boiled for 10 min at 95°C. After measuring the total protein concentration of each sample with the NI-protein assay (G-Biosciences), an equivalent amount of total protein for each sample was then separated on a 10% SDS-PAGE gels. Bocillin-labeled proteins were then imaged using a Typhoon 9500 fluorescence imager (GE Healthcare) with excitation at 488 nm and emission at 530 nm.

## Image acquisition and analysis

Growth conditions and staining procedures prior to microscopy are described in the figure legends. Prior to imaging, cells were immobilized on 2% agarose pads containing the appropriate growth medium, and covered with #1.5 coverslips. Images were cropped and adjusted using FIJI software.

Micrographs were obtained using a Nikon Ti inverted microscope outfitted with a Nikon motorized stage with an OkoLab gas incubator with a slide insert attachment, an Andor Zyla 4.2 Plus sCMOS camera, Lumencore SpectraX LED Illumination, Plan Apo lambda 100x/1.45 Oil Ph3 DM objective lens, and Nikon Elements 4.30 acquisition software. Images in the green and red channels were taken using Chroma 49002 and 49008 filter cubes, respectively. The microscope was maintained at 30℃ using a custom-made environmental control chamber.

For the demographs, a custom-written MATLAB code, cells were arranged from top to bottom according to their cell lengths. Additionally, each cell was oriented such that the cell pole with the higher bait intensity are located on the right. The demograph of the prey signal was plotted using this same cell orientation.

## Phylogenetic tree generation

To generate the phylogenetic tree showing the distribution of DUF 3556 containing proteins in a diverse set of bacterial taxa, the amino acid sequences of CofA was submitted to the NCBI conserved domain search (*Marchler-Bauer et al., 2017*). NCBI reported 13104 proteins that contain the DUF 3556 (found using RPS BLAST), which were utilized for the phylogenetic tree. We used a complex and diverse set of 1773 bacterial taxa called 'Representative Genomes' that is available on NCBI (ftp://ftp.ncbi.nlm.nih.gov/blast/db/, Representative_Genomes.00.tar.gz). The phylogenetic tree was constructed using PhyloT (http://phylot.biobyte.de/) and BLASTp results were plotted against the tree. The tree was visualized and annotated using iToL (http://itol.embl.de/) (*Letunic and Bork, 2016*).

## Acknowledgements

The authors gratefully acknowledge all members of the Bernhardt and Rudner labs for advice and helpful discussions as well as the microscopy services provided by Paula Montero Llopis and her team at the Microscopy Resources on the North Quad (MicRoN) core facility at HMS. This work was supported by a HHMI-Simons Faculty Scholar award to T.G.B., the National Institutes of Health (AI083365 to T.G.B), and Investigator funds from the Howard Hughes Medical Institute. H.C.L. was supported in part by the Life Sciences Research Foundation, where he was a Simons Fellow. J.W.S. was supported in part by the T32 Bacteriology PhD Training Program (AI132120-02) awarded to the Harvard Graduate Program in Bacteriology.

## Additional information

### Funding

| Funder | Grant reference number | Author |
|---|---|---|
| National Institute of Allergy and Infectious Diseases | AI083365 | Thomas G Bernhardt |
| National Institute of Allergy and Infectious Diseases | AI132120 | Joel W Sher |
| Life Sciences Research Foundation | | Hoong Chuin Lim |
| Howard Hughes Medical Institute | Faculty Scholar Award | Thomas G Bernhardt |
| Simons Foundation | Faculty Scholar Award | Thomas G Bernhardt |
| Howard Hughes Medical Institute | Investigator funds | Thomas G Bernhardt |

The funders had no role in study design, data collection and interpretation, or the decision to submit the work for publication.

## Author contributions

Joel W Sher, Conceptualization, Investigation, Methodology, Writing - original draft, Writing - review and editing; Hoong Chuin Lim, Conceptualization, Resources, Supervision, Writing - original draft, Writing - review and editing; Thomas G Bernhardt, Conceptualization, Supervision, Funding acquisition, Writing - original draft, Writing - review and editing

## Author ORCIDs

Joel W Sher (iD) https://orcid.org/0000-0002-0616-7632
Hoong Chuin Lim (iD) https://orcid.org/0000-0001-8463-8375
Thomas G Bernhardt (iD) https://orcid.org/0000-0003-3566-7756

## Decision letter and Author response

Decision letter https://doi.org/10.7554/eLife.54761.sa1
Author response https://doi.org/10.7554/eLife.54761.sa2

# Additional files

## Supplementary files

• Supplementary file 1. Growth conditions for the phenotypic profiling. The different growth conditions used for the profiling analysis are listed. Samples 1 g_A, B, and C, correspond to those sequenced to analyze the transposon insertion profile in the original library following one generation of growth. Similarly, 11 g_A and 11 g_B correspond to samples grown for 11 generations without treatment.

• Supplementary file 2. Plasmid construction methods.

• Transparent reporting form

## Data availability

Raw sequencing data is available at Sequence Read Archive (BioProject ID: PRJNA610521, http://www.ncbi.nlm.nih.gov/bioproject/610521). All codes referenced in the manuscript are available at https://github.com/jsher-Bernhardtlab/cofA (copy archived at https://github.com/elifesciences-publications/cofA). All other relevant data are within the manuscript and supplementary files.

The following dataset was generated:

| Author(s) | Year | Dataset title | Dataset URL | Database and Identifier |
|---|---|---|---|---|
| Sher JW, Lim HC, Bernhardt TG | 2020 | Phenotypic profiling of a Corynebacterium glutamicum transposon library | https://www.ncbi.nlm.nih.gov/bioproject/PRJNA610521 | NCBI BioProject, PRJNA610521 |

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
