## [Decision Letter]

**Acceptance summary:**

Processes associated with bacterial elongation, cytokinesis and cell separation represent attractive avenues for the development of new antimicrobial agents with novel modes of action. These pathways have been well studied in model gram positive and gram negative organisms, but remain poorly understood in actinobacteria and other atypical bacteria. This study uses global phenotypic profiling of a high density transposon library of *Corynebacterium glutamicum* to identify CofA, a small transmembrane protein that is required for stable polar localization of Pbp1a. The work represents an important advance in the understanding of how the cell wall biosynthetic machinery is coordinated and stabilized in actinobacteria. The resulting hits from the transposon screen conducted in the presence of antibiotics represent an important resource for the field and will assist in advancing this important area of research.

**Decision letter after peer review:**

Thank you for submitting your article "Global phenotypic profiling identifies a conserved actinobacterial cofactor for a bifunctional cell wall synthase" for consideration by *eLife*. Your article has been reviewed by three peer reviewers, one of whom is a member of our Board of Reviewing Editors, and the evaluation has been overseen by Gisela Storz as the Senior Editor. The following individual involved in review of your submission has agreed to reveal their identity: Neeraj Dhar (Reviewer #2).

The reviewers have discussed the reviews with one another and the Reviewing Editor has drafted this decision to help you prepare a revised submission.

Summary:

There is a growing body of evidence suggesting that bacterial elongation, cytokinesis and cell separation in actinobacteria are distinct from other well-studied model organisms. Elongation of cells by incorporation of new cell wall subunits at the cell tip in mycobacteria has been reported to occur through the activity of a tropomyosin like protein, DivIVA. It has been demonstrated that the phosphorylation state of DivIVA is important for protein-interactions however, further mechanistic insight on how this protein co-ordinates the recruitment of other biosynthetic enzymes/machineries is lacking. Further, there is a relative paucity of information on the identity of other protein complexes, if any, that assemble at the cell tip during elongation. In this study, the authors use a transposon library in another actinobacterial species, Corynebacterium glutamicum, exposed to various antibiotics to identify genes required for cell wall assembly and cell tip growth.

Key findings:

1) The authors grew a transposon mutant library of C. glutamicum in the presence of antibiotics that target different processes and measured fitness scores of mutants by comparing prevalence of DNA when bacteria are grown with and without antibiotic. They report that genes with similar function, or those that function together in the cell (or have similar responses to antibiotic), cluster together as expected. They were also able to ascribe new functions to some genes that cluster together with other known functional classes.

2) Whilst numerous interesting hits emerged from the screen, the authors decided to focus further effort on a transmembrane protein, CofA (co-factor of PBP1a), which displayed a similar phenoprint to ponA (PBP1a), a high molecular weight penicillin binding protein. Deletion of cofA resulted in increased sensitivity to ampicillin and meropenem.

3) The authors demonstrate that CofA is required for stabilization of PBP1a using fluorescent fusions of both proteins and bocillin.

4) Using their previously reported PopZ-Linked Apical Recruitment two-hybrid assay, the authors report that CofA interacts with PBP1a via its transmembrane domain and is also able to interact with itself.

5) Domain swapping of the transmembrane domain of C. glutamicum PBP1A and with that of *E. coli* abrogated dependency of PBP1a on CofA for stabilization. However, recruitment of PBP1a to the cell tip and interaction with CofA was affected suggesting that the transmembrane domain is required for CofA interaction and stable polar localization of PBP1a

6) Using CofA paralogs from Corynebacterium jekium and *Mycobacterium tuberculosis*, the authors confirm that a similar phenomenon may be occurring in other actinobacteria. The *M. tuberculosis* CofA-PBP1a complex required deletion of an extended N-terminal component.

This is an important description of a conserved actinobacterial PBP interacting protein that modulates PBP function/localization. This work is an extremely helpful resource for the community and an important step forward for the identification of unknown cell wall acting factors.

Essential revisions:

1) Figure 1: It would be good to show the growth curves in a supplementary figure to give readers an idea about the degree of growth retardation in presence of the antibiotics/stresses.

2) The that data shown in Figure 3 appears promising but incomplete. The data suggests that the genes are probably involved in undecaprenol synthesis or utilization. Since this is not the main focus of the study and is just an example of the phenotypic profiling approach, this can be moved to the supplementary data.

3) Figure 4D – microscopy should be quantitated in some way. How many cells were viewed in total etc? There does appear to be some residual PBP1a localization, please comment. Similarly, Figure 1E (and Figure 6A) should have some quantitation of the bands.

4) The authors make the statement, "Furthermore, the reduced severity of the phenotypes for CofA inactivation relative to PBP1a is consistent with a model in which CofA is required to promote PBP1a activity but that PBP1a retains some residual function in the absence of CofA." The evidence base for this is unclear, please explain further.

---

## [Author Response]

Essential revisions:1) Figure 1: It would be good to show the growth curves in a supplementary figure to give readers an idea about the degree of growth retardation in presence of the antibiotics/stresses.

Example growth curves of *Cglu* treated with two different drugs (Vancomycin or Erythromycin) have been added in Figure 1—figure supplement 1 to illustrate the effect the selected drug concentrations on *Cglu* growth rate. Also, the time it took the standard inoculum of the *Cglu* transposon library to reach an OD_600_ of 0.5 for each treatment condition was added to Supplementary file 1. Lastly, the measured MIC for each drug on *Cglu* was added to Supplementary file 1. These additions should give the reader a good sense of the growth retardation experienced by *Cglu* under each condition used for the analysis.

2) The that data shown in Figure 3 appears promising but incomplete. The data suggests that the genes are probably involved in undecaprenol synthesis or utilization. Since this is not the main focus of the study and is just an example of the phenotypic profiling approach, this can be moved to the supplementary data.

As recommended, the figure has been moved to the supplement (Figure 2—figure supplement 1).

3) Figure 4D – microscopy should be quantitated in some way. How many cells were viewed in total etc? There does appear to be some residual PBP1a localization, please comment. Similarly, Figure 1E (and Figure 6A) should have some quantitation of the bands.

Demographs representing the localization of fluorescent protein fusions for each sample in >345 cells for each strain is now presented in Figure 3—figure supplement 1D. In the micrographs in Figure 3D there does seem to be a slight residual signal of mScar-PBP1a at the septa of ΔcofA cells. However, this may simply be due to the double membrane at septa that slightly raises the signal of the membrane protein fusion above background. No significant signal was observed in the demographs provided. As requested, the fold change for the relevant bands was quantified and added in the legends for Figure 3E and Figure 5A.

4) The authors make the statement, "Furthermore, the reduced severity of the phenotypes for CofA inactivation relative to PBP1a is consistent with a model in which CofA is required to promote PBP1a activity but that PBP1a retains some residual function in the absence of CofA." The evidence base for this is unclear, please explain further.

This statement was revised to read, “However, the reduced severity of the phenotypes for CofA inactivation relative to PBP1a inactivation suggests that PBP1a retains some residual function in the absence of CofA. This phenotypic disparity is consistent with a model in which CofA promotes normal PBP1a function but is not absolutely required for its activity.”